# Recent Progress in Self-Powered Sensors Based on Liquid–Solid Triboelectric Nanogenerators

**DOI:** 10.3390/s23135888

**Published:** 2023-06-25

**Authors:** Quang Tan Nguyen, Duy Linh Vu, Chau Duy Le, Kyoung Kwan Ahn

**Affiliations:** 1Graduate School of Mechanical Engineering, University of Ulsan, Daehakro 93, Nam-gu, Ulsan 44610, Republic of Korea; paxnguyen91@ulsan.ac.kr; 2School of Mechanical Engineering, University of Ulsan, Daehakro 93, Nam-gu, Ulsan 44610, Republic of Korea; dlinh5832@ulsan.ac.kr; 3Faculty of Electrical and Electronic Engineering, Ho Chi Minh City University of Technology (HCMUT), 268 Ly Thuong Kiet Street, District 10, Ho Chi Minh City 700000, Vietnam; lechauduy@hcmut.edu.vn; 4Vietnam National University Ho Chi MInh City, Linh Trung Ward, Ho Chi Minh City 700000, Vietnam

**Keywords:** self-powered sensor, flexibility sensor, triboelectric nanogenerator, liquid–solid interface, active sensor, chemical sensor

## Abstract

Recently, there has been a growing need for sensors that can operate autonomously without requiring an external power source. This is especially important in applications where conventional power sources, such as batteries, are impractical or difficult to replace. Self-powered sensors have emerged as a promising solution to this challenge, offering a range of benefits such as low cost, high stability, and environmental friendliness. One of the most promising self-powered sensor technologies is the L–S TENG, which stands for liquid–solid triboelectric nanogenerator. This technology works by harnessing the mechanical energy generated by external stimuli such as pressure, touch, or vibration, and converting it into electrical energy that can be used to power sensors and other electronic devices. Therefore, self-powered sensors based on L–S TENGs—which provide numerous benefits such as rapid responses, portability, cost-effectiveness, and miniaturization—are critical for increasing living standards and optimizing industrial processes. In this review paper, the working principle with three basic modes is first briefly introduced. After that, the parameters that affect L–S TENGs are reviewed based on the properties of the liquid and solid phases. With different working principles, L–S TENGs have been used to design many structures that function as self-powered sensors for pressure/force change, liquid flow motion, concentration, and chemical detection or biochemical sensing. Moreover, the continuous output signal of a TENG plays an important role in the functioning of real-time sensors that is vital for the growth of the Internet of Things.

## 1. Introduction

Over the past few years, the challenges posed by climate change and energy shortages have become increasingly extreme, thereby highlighting the pressing need for clean and renewable energy [1]. Despite the vast amounts of energy contained within human footfalls, ocean waves, raindrops, and airflow, a significant portion of it is wasted due to the difficulty of harnessing it effectively [2,3]. Therefore, it is crucial to develop suitable technology to tap into this energy source, and advancements in nanotechnology have led to the creation of various nanogenerators for harvesting mechanical energy [4,5,6]. Wang and his team developed triboelectric nanogenerators (TENGs) which use the electrostatic and triboelectrification effects to turn mechanical energy into electricity [7]. Triboelectrification, also known as contact electrification, occurs when two materials come into physical contact and become electrically charged due to friction. The sign of generated charges at the contact interface depends on the relative polarities of the materials involved. One material tends to lose electrons, while the other material tends to gain electrons, resulting in the generation of positive and negative triboelectric charges, respectively. TENGs are capable of generating electricity from various energy sources, including vibration, wind, wave water, and human motion, among others [8,9,10,11]. Since their inception, researchers have studied various TENG structures and functions to enhance their output performance and energy conversion efficiency [12,13,14,15].

Aside from the extraordinarily rapid development of the solid–solid TENG [16,17,18], the liquid–solid TENG (L–S TENG) has also become a potential trend due to its stable output and durability [19,20,21,22,23,24]. The L–S TENG device has been developed based on a simple concept: that water drops make contact with the dielectric solid surface and generate electricity. For example, a metal liquid was used for the TENG device and could harvest energy with a high conversion efficiency of 70.6% [25]. Energy from waves has been recovered by using the flexible L–S TENG; indeed, it is considered a power supply solution anywhere as long as wave energy is available [26]. In addition, electricity has been generated from L–S TENG devices when water has flown through elastic silicon tubing [27]. Despite the lack of complete understanding regarding the specific electrostatic effect, attempts have been made to elucidate the operational concept behind the interaction between liquid and solid by considering the motion of triboelectric charge on the material’s surface.

The L–S TENG can be used in specific applications depending on the type of liquid–solid contact. A collection of photographs of the L–S TENG used for energy harvesting and self-powered sensors is shown in Figure 1 [28,29,30,31,32,33]. Energy harvesting from tidal and oceanic waves, rainfall, and water streams offers enormous promise for the L–S TENG. The L–S TENG gadget has been successfully used in self-powered sensors in recent years because of its capacity to transform mechanical energy into electric output without the need for a power unit. As a result, this review concentrates on the most current developments in self-powered sensors based on the L–S TENG principle in addition to exploring their practical applications. The review begins with a description of the basic operation of an L–S TENG and various operating modes, followed by a summary of the variables that affect it. Secondly, a range of self-powered sensors based on the L–S TENG—such as active pressure/touch sensors, [31] chemical sensors [34], biological sensors [35], gas sensors [36,37], and so on—will be summarized as strong applications for TENG devices. Finally, the conclusion section highlights future opportunities and perspectives for the development of self-powered sensors based on the L–S TENG.

## 2. Liquid–Solid Contact Triboelectric Nanogenerator

### 2.1. Mechanism of Liquid–Solid Contact Electrification

Contact electrification (CE) is a fundamental phenomenon in electricity generation that occurs when two materials are physically in contact with each other, generating friction [38]. The exact mechanism behind the transfer of charges has been a topic of debate for many years, without no definitive conclusion having been reached. In an attempt to explain CE at the atomic level, Wang et al. [39] proposed the interatomic interaction model, also called the Wang transition. Accordingly, the dominant mechanism for CE is suggested to be electron transfer. As depicted in Figure 2a, CE can occur at contact interfaces between solids, liquids, and gases.

When two atoms form a bond, an equilibrium interatomic distance is established. If the distance is shortened by an externally applied force, the atoms will repel each other due to increased electron cloud overlap. Conversely, if the distance is longer, the atoms will attract due to decreased overlap, as shown in Figure 2b. Furthermore, Wang et al. [39] found that CE between two surfaces only occurs due to the electron transfer from one surface to the other when the interatomic distance between contacting points is forced shorter than the equilibrium distance. Subsequently, upon the separation, the transferred electrons remain at the contacting points as stationary charges on the material’s surface. The electrostatic charges are released through thermal ionic emission or photon excitation. The Wang transition model explains the occurrence of liquid–solid CE, which happens when liquid molecules collide with solid surface atoms under liquid pressure, leading to electron transfer and electron cloud overlap [40]. To further understand this process, Wang et al. [39] proposed a “two-step” model for the formation of the electric double layer (EDL), with electron transfer playing a dominant role in the first step. As shown in Figure 2c, when a liquid makes contact with a solid surface, electron cloud overlap can result in electron transfer and charge the solid surface. Under liquid flow pressure, ions attach to the surface, forming a layer with freely migrating ions in the liquid. Positive ions, attracted by electrostatic interactions, migrate toward the surface and form an EDL in the second step. CE due to electron transfer likely initiates the formation of the EDL [41].

### 2.2. Basic Mode of Operation of Liquid–Solid Triboelectric Nanogenerator

A liquid–solid triboelectric nanogenerator (L–S TENG) is a highly efficient device capable of converting different types of mechanical energy into electrical energy, with a particular focus on applications involving flowing water. This is achieved through the coupling effects of liquid–solid CE and electrostatic induction. Previous research has identified four distinct modes of operation for the L–S TENG, which are determined by the movement of the triboelectric layer and the configuration of the electrodes. These modes include contact-separation [42,43,44,45,46], lateral sliding [47,48,49], free-standing [19,20,50,51,52,53,54,55,56,57,58,59,60,61,62,63], and single-electrode [15,33,64,65,66,67,68,69,70,71,72,73] modes.

#### 2.2.1. Contact-Separation Mode

In the contact-separation mode of the L–S TENG, electrical energy is produced through electrical induction, which occurs when a solid surface oscillates vertically and makes contact with water before separating from it [42,43,44,45,46]. As can be seen in Figure 3a, Lin et al. [43] reported on a water-TENG based on this principle, which achieved a voltage of 52 V, and current and power densities of about 2.45 mA/m^2^ and 0.13 W/m^2^, respectively. Before contact, the polydimethylsiloxane (PDMS) surface remains uncharged, and no charge transfer occurs between PDMS and water (Figure 3(ai)). However, when the PDMS surface comes into contact with water under an external pressing force, CE occurs, generating a negatively charged PDMS surface and a positively charged EDL (Figure 3(aii)). Once the PDMS separates from the water, the transferred charges remain on the PDMS surface, breaking the electrical neutrality and establishing a difference in electric potential. This leads to inducing positive and negative charges on the corresponding electrodes to balance the potential difference. As a result, electrons flow from Cu electrode 2 to Cu electrode 1, generating a current in the external circuit (Figure 3(aiii)). The current output obtains the maximum when PDMS returns to the initial position (Figure 3(aiv)). When the PDMS is forced to move closer and then make contact again with the water surface, the potential difference decreases, causing a reverse flow of electrons and generating a current in the opposite orientation (Figure 3(av)). Once a new equilibrium state is established, the device returns to its initial state, and no current is generated. Due to the periodic contact separation of PDMS and water, the TENG provides a continuous output through the external circuit with alternating current characteristics.

Similarly, a liquid–solid-based triboelectric generator (LSTEG) was proposed by Yang et al. [46], and can produce a significant amount of power, with a power density of 9.62 W/m^2^. The working mechanism of the LSTEG is illustrated in Figure 3b, where contact between a PTFE and water generates triboelectric charges on their surfaces (Figure 3(bii)). As the PTFE separates from the water surface, the negatively charged PTFE surface induces positive charges on Cu electrode 1, while negative charges are induced on Cu electrode 2 (Figure 3(biii)), generating a positive current flow through the external circuit until the PTFE returns to the initial position (Figure 3b(iv)). When the PTFE is pressed into contact again with the water, a reverse current flow occurs until a new state of equilibrium is reached (Figure 3(bv)). It is clear that one press–release cycle of the PTFE can produce an alternating current.

#### 2.2.2. Lateral Sliding Mode

The lateral sliding mode is another power generation mode that utilizes the same structure as the contact separation mode but with a different manner of movement. Instead of relying on contact separation, it produces power through lateral movement between two contact surfaces, where relative friction plays a crucial role [47,48,49]. The working mechanism of a lateral sliding mode L–S TENG is illustrated in Figure 4a. Nahian et al. [48] proposed a lateral-sliding-style fluid-based triboelectric nanogenerator (L–S FluTENG), which consists of an aluminum tape covered by a PTFE layer and is located on the outer surface of a tube, and a cylindrical reservoir with a Cu located on the inside surface. Initially, the PTFE and water surfaces are uncharged (Figure 4(ai)). When the PTFE slides are immersed in the water, CE occurs at the contact interface, resulting in positively and negatively charged water and PTFE surfaces, respectively (Figure 4(aii)). Next, once the PTFE emerges from the water, the charge balance is broken, and electrons are attracted to flow from the aluminum electrode to the copper electrode to neutralize the unbalanced charges, producing a current through the external circuit until completely separate from the water (Figure 4(aiii)). Subsequently, when the PTFE slides are immersed again in the water, the positive water forms an interfacial EDL with the PTFE (Figure 4(aiv)). Electrons will flow inversely to neutralize the charged electrodes. This figure also displays the voltage and current outputs of the LS-FluTENG, which proves its AC characteristics, with a peak voltage of 6 V and peak current of 300 nA. In the contact-separation mode of the L–S TENG, the electrical energy is produced via electrical induction.

In addition, Lee et al. [47] reported a water-based TENG, in which an Al plate covered by Teflon AF 1600 (PTFE) periodic slides are either immersed into the water or emerge out of the water, as depicted in Figure 4b. When the PTFE and water are in physical contact, CE occurs, leading to the generation of negatively and positively charged PTFE and water surfaces, respectively. Once the PTFE emerges from the water, charged PTFE and water surfaces induce positive and negative charges on the Al and Cu electrodes, leading to generating a current flow through the external circuit, and maximizing when the PTFE is completely out of the water. When the PTFE immerses in the water again, a current flow with an inverse direction is generated in the external circuit. Therefore, the periodic slides that are either immersed into the water or that emerge out of the water can produce alternating currents. This is confirmed experimentally by the voltage and current outputs generated by the WTENG at a movement frequency of 2 Hz.

#### 2.2.3. Free-Standing Mode

The typical free-standing mode of the TENG involves a free-moving triboelectric object and two electrodes [19,20,50,51,52,53,54,55,56,57,58,59,60,61,62,63]. For instance, Kim et al. reported a rotating water TENG that can generate a maximum instantaneous open-circuit voltage of 27.2 V and a short-circuit current of 3.84 µA [56]. The step-by-step mechanism of the device is illustrated in Figure 5a. This device consists of PTFE-coated Al electrodes positioned on the inner surface of a cylinder, which is partially filled with water. The CE occurring at the contact interface of the water and PTFE leads to producing AC outputs. During the rotation, the water gradually deforms under the effect of gravity and dynamic inertia. Furthermore, electrodes E1 and E2 located on the opposite side of the cylinder are connected. In stage (i), E1 and E2 establish an electrical equilibrium state. As E1 emerges out of the water and E2 makes contact with the water due to the rotation of the cylinder, the equilibrium state is disrupted, creating an unbalance potential between the water and PTFE, leading to current flow through the external circuit. Once E2 fully makes contact with the water, a new state of equilibrium forms. CE continuously occurs between the water and PTFE, and the electrostatic induction process induces an inverse current flow in the external circuit. Both the voltage and current outputs exhibit AC characteristics. Another example is the U-shaped TENG constructed by Zhang et al. [20]. This device comprises an FEP U-shaped tube, with two Cu electrodes located on the outer surface of two columns of the FEP U-shaped tube, partially filled water (Figure 5b). The continuous up-and-down flow of water along the left and right columns generates electrostatic charges due to the CE of water and FEP, leading to electrical energy generation via electrostatic induction on two electrodes. The device can produce a peak voltage of 20 V and a peak current of 400 nA.

#### 2.2.4. Single-Electrode Mode

Another important mode of operation of the L–S TENG is the single-electrode mode that generates electricity through a single electrode connected to the ground. Compared to the other modes—such as the contact-separation, lateral sliding, and free-standing modes—which generate electricity via two electrostatic induction processes between two electrodes, a single-electrode-based L–S TENG has several advantages [15,33,64,65,66,67,68,69,70,71,72,73].

Lin et al. demonstrated a water-TENG with a superhydrophobic PTFE surface, working in a single-electrode mode, that can convert the energy from flowing water and falling droplets (Figure 6a) [15]. This device can generate a maximum voltage and current of 9.3 V and 17 µA, respectively, from a water droplet of about 30 µL. When a water droplet falls onto the surface of a PTFE thin film, CE occurs at the water–PTFE interface, resulting in the generation of a negatively charged PTFE surface and a positive EDL, and an electrical equilibrium is formed. Once the water droplet spreads out from the PTFE, the electrical equilibrium is broken, leading to the development of a potential difference across the Cu electrode and the ground, driving the current flow in the external circuit. Electrons from the Cu electrode flow to the ground until reaching equilibrium, generating a negative current. When other water droplets make contact with the PTFE surface, the negatively charged PTFE surface attracts the positive ions inside the droplet to form an EDL, establishing unbalance potential between the Cu electrode and the ground. Electrons are attracted to flow from the ground, generating a positive current through the external circuit. In the case that the droplet leaves the PTFE film, an inverse current is obtained until another new equilibrium is achieved. Consequently, the periodic falling of water droplets exhibits a continuous AC output. Yang et al. also presented a similar study in which a water-droplet-driven TENG (Wd-TENG) demonstrated that positive and negative current peaks can be generated during the contact separation of a falling droplet and a triboelectric layer surface [66]. The working mechanism of the Wd-TENG and its typical current signal concerning the droplet’s motion is shown in Figure 6b.

### 2.3. Interacting Modes of Liquid–Solid Triboelectric Nanogenerator

The L–S TENG can be regarded as a promising technology for transforming mechanical energy from liquid flow into electricity. Based on the source of the liquid dynamics, the L–S TENG can be divided into three main modes: the droplet [15,42,45,53,54,55,60,61,63,64,66,67,68,69,71,72,74,75,76,77,78,79,80], wave [33,44,49,50,51,52,56,59,65,81,82,83], and flow-based [19,20,58,62,73,84] modes. The most important source for L–S TENGs is water, which exists everywhere in nature in enormous amounts, as seen with raindrops, rivers, tides, and ocean waves. Therefore, the interaction modes of the L–S TENG will be reported in the following sections based on the water-TENG.

#### 2.3.1. Droplet-Based L–S TENG

Droplet-based mechanisms are commonly used in L–S TENGs to harvest energy from falling droplets or raindrops, which interact with a charge-generating layer and induce charges on the electrodes and generate a current in an external circuit. The charge-generating layer could be an insulator [15,42,45,53,54,55,60,61,64,66,67,68,69,71,72,76,77,78,79,80], semiconductor [23,63,75,85,86,87], or conductor [74] material. Lee et al. proposed a water-droplet-driven TENG (WdTENG) that works in the contact-separation mode, where a water droplet bounces between two superhydrophobic surfaces made by a typical negative triboelectric material, PTFE, as shown in Figure 7a [45]. This device can generate a peak voltage and current of about 6.8 V and 6 Μa, respectively, measured at an inclination angle of 70°. Lu et al. reported an L–S TENG that converts energy from the movement of a water droplet between two semiconductors with different Fermi levels (Figure 7b) [63]. This device exhibits a peak voltage and current of 0.3 V and 0.64 μA, respectively, with DC characteristics. Unlike conventional TENGs, which normally produce AC outputs, this study provided a new approach for harvesting low-frequency vibration energy directly into DC power. It also highlighted the potential for integrated and miniaturized generators, as the output characteristics of the device can be determined by relative parameters such as the speed, direction, and volume of the water droplet.

Moreover, for enhancing the output performance, various affecting parameters have been investigated, including the selection of triboelectric materials and the device concept. Traditionally, the droplet-based mechanism uses one or two electrodes covered by an insulator, where charges generated by the friction energy of water and the triboelectric layer remain on the solid surface after the separation of water from the layer. This induces the electrostatic charges to flow between two electrodes (double-electrode mode) or between one electrode and the ground (single-electrode mode). Accordingly, these devices typically generate AC electricity [15,42,54,55,60,66,69,79].

Recently, a new methodology for electricity generation has emerged based on a new electrode structure design. In this design, the charge generated due to CE between water and the triboelectric layer will directly transfer to the electrode when the charge-carried water droplet makes direct contact with the electrode [67,68,72,76,77,78,80]. This leads to a different electrical response compared to traditional droplet-based TENGs. For example, Xu et al. proposed a droplet-based device using PTFE with a tiny piece of aluminum deposited onto a glass substrate with an ITO electrode underneath [80]. This device can output high values of instantaneous voltage and current—143.5 V and 270.0 μA, respectively, to be exact—which are much higher than those generated when not using an Al electrode. Wang et al. reported a similar device (SHS-DEG) consisting of structurally hierarchical and superhydrophobic FEP with a Cu electrode underneath the FEP film and an aluminum electrode located on the FEP surface (Figure 7c) [68]. SHS-DEG can produce a peak voltage of about 200 V and a peak current of 400 μA.

#### 2.3.2. Flow-Based L–S TENG

Another important interacting mode of the L–S TENG is the continuous CE that occurs at the contact interface of a streaming-flow liquid and a solid surface [19,20,58,62,73,84]. This mode of interaction allows for the conversion of mechanical energy from the flow of water into electrostatic energy. This electrostatic energy is then converted into electricity through either electrostatic induction [19,20,58,62,73] or the breakdown effect [84]. One example of this is the SWING stick developed by Choi et al., which converts the mechanical energy from shaking into electricity (Figure 7d) [62]. The stick generates electricity through CE between the water and Teflon, creating positive charges inside the water and negative charges on the Teflon surface. When the charged water makes contact with the bare Al tube, the generated charges inside the water are neutralized, generating a current through the external circuit. Other examples include the U-tube TENG developed by Zhang et al. [20] and Pan et al. [19] (Figure 7e), which can generate a stable peak output voltage and current of about 20 V–400 nA, and 350 V–1.75 μA, respectively [20]. These devices can also function as multifunctional sensors, such as displacement and pressure sensors, with high sensitivity. Another example is the PTFE–copper (PCTENG) tube developed by Munirathinam et al. (Figure 7f) [73], which can harvest energy from flowing water and achieve voltage, current, and power peaks of 36 V, 0.8 µA, and 45 µW, respectively. The electricity is generated through CE occurring between the flowing water and the PTFE tube.

#### 2.3.3. Wave-Based L–S TENG

Compared with the droplet- and flow-based L–S TENG, the wave-based L–S TENG has also received significant attention from scientists due to the large amount of energy contained in dynamic waves. In a recent study, Zhu et al. [51] reported the development of an LSEG using an FEP thin film with an array of electrodes underneath (Figure 7g). The LSEG generates AC electricity by combining triboelectrification and electrostatic induction during traveling water waves. This study demonstrated the use of the LSEG as an effective energy harvester from ambient water waves and also showed the potential of the LSEG in both onshore and offshore areas, as well as rainy area applications, contributing an important and promising solution for the sustainable development of society. Li et al. designed a buoy L–S TENG that can output a high current and voltage of 290 µA and 300 V, respectively, by synchronizing the outputs of a network of 18 L–S TENGs (Figure 7h) [65]. The produced energy is suitable to power a wireless SOS system for ocean emergencies. Importantly, the wave-based L–S TENG also demonstrated its potential to be used in self-powered sensor systems. Xu et al. developed an L–S TENG that can be used as a wave height sensor for smart marine equipment (Figure 7i) [33].

## 3. Affecting Parameter on L–S TENG Performance

Studying the affecting parameters of L–S TENGs is a complex and challenging task due to the diverse device designs and working modes involved. Nonetheless, ongoing research is actively addressing this challenge by focusing on the factors that impact the output performance and durability of L–S TENG devices. The liquid phase and solid phase are the two major groups into which these influencing factors can be divided. Liquid phase parameters include elements such as the type of liquid, viscosity, and surface tension, all of which can affect how well the L–S TENG performs. Meanwhile, solid phase properties refer to the type of solid material used, surface morphology, and structure shape, which can also affect the output of these devices. Therefore, understanding these affecting parameters and developing methods to optimize the output performance and durability of L–S TENGs are essential for their widespread adoption. Despite the challenges associated with studying L–S TENGs, the potential benefits of this technology make it a promising field of research. By gaining a better understanding of the parameters affecting L–S TENGs, researchers can work towards developing more efficient and durable devices that can be utilized in a broad range of applications.

The solid phase of an L–S TENG consists of two essential components: the contact layer and the electrode layer. The first consideration is the electrode layer, as the choice of materials for this layer is critical to the performance of the device. The selected material with high electrical conductivity can significantly enhance the electronic transfer from the contact layer, leading to a more efficient output performance. In addition to conductivity, flexibility is also an important factor when fabricating TENG devices with various models. Over the last few years, several materials, including aluminum, silver, gold, and copper, have been commonly used as electrode materials in L–S TENG devices. These materials possess high conductivity, good flexibility, commercial availability, and well-researched properties, making them ideal for TENG fabrication. Therefore, selecting the appropriate electrode material is crucial in the design and performance of L–S TENG devices. Through careful selection and utilization of materials, researchers can continue to improve the output performance and durability of TENGs and expand their potential applications [88,89,90,91]. Additionally, numerous other conductive materials have been suggested owing to their shared characteristics of flexibility, stretchability, and notable chemical stability. These materials include carbon nanotubes (CNTs) [92,93], graphene [94], nanowire-based materials [95], as well as organic or polymer-based materials [96].

In addition to the charge-collecting layer, the material charge density and hydrophobicity play crucial roles in enhancing the output of TENG. It is essential for the contact layer material to possess a high negative charge due to the positive triboelectric properties of the liquid. To quantitatively evaluate the triboelectric effect and establish a standard, the triboelectric charge density (TECD) was measured to rank different materials (Figure 8) [97]. In a triboelectric series, some common materials have been used with low TECD, such as PVC, PTFE, PDMS, Kapton, and PVDF, equivalent to TECD values −117.5, −113.1, −102.2, −92.9, and −87.4 mC·m^−2^, respectively [97]. Moreover, the negative charge of the material can be enhanced through the utilization of an air-ionization gun, which induces corona discharging of the surrounding air. By employing corona discharging, the TECD can be increased by more than 5 times compared to the initial material [98]. Extensive research has also been conducted on improving output performance through the use of hydrophobic surfaces. Various methods have been employed to create highly hydrophobic surfaces, including the fabrication of nanostructures or hierarchical structures [71], artificial lotus-leaf structures [53], and plasma treatment [99]. It is noteworthy that the surface morphology of the contact layer affects the velocity of the liquid on the contact layer and the bouncing motion between the liquid–solid surface. The hydrophobic surface is characterized by contact and a sliding angle. High contact and a high sliding angle will increase the current output in the droplet single-electrode contact mode [45].

When considering the liquid-phase properties of the L–S TENG, two main types of liquid have been utilized: metal liquid and water. Metal liquids such as mercury and Galinstan have been chosen due to their liquid state at room temperature, excellent fluidity, and conductivity. However, their effect on TENG devices has not been studied in any paper, and they are typically used as a replacement for solid metals, acting as an electrode layer [25,100,101]. On the other hand, the properties of water, carefully studied, include the water forms (droplet, waves, flow), ion type, and concentration of other materials soluble in water. It is well known that the water form has a significant impact on the amount and frequency of contact between liquid and solid surfaces. With a larger contact area and higher frequency, the output power will obtain a higher result. The droplet water has been investigated to find out the effect of droplet volume, falling height, and tilting angle on L–S TENG output performance [102]. It can be seen that a droplet’s volume is proportional to the velocity of the droplet. Therefore, the inertial force is affected by the droplet volume and can be expressed as $\rho v^{2}/D$, where ρ is density, *v* is velocity, and *D* is the diameter of the droplet. However, the inertial force when the droplet moves down on the solid surface is still affected by the velocity of the droplet increase in time ($\vartheta_{t}$) due to the falling height (*h*) and tilting angle (θ). This relationship is expressed using Equation (1):(1)$$\vartheta_{t}=\sqrt{2gh}sin\theta$$

Due to the increment of the kinetic energy, $\vartheta_{t}$ increases when falling height (*h*) increases, leading to an increase in the current output. Likewise, higher θ is attributed to the increase in $\vartheta_{t}$ in Equation (1). However, the current output reaches saturation when the angle exceeds 45° and then drops when the inclination angle is over 75°. Additionally, the number of transferred charges produced at the interface between a droplet and a solid is closely linked to the velocity at which the droplet slides across the hydrophobic surface. Specifically, higher sliding velocities result in a greater quantity of transferred charges [103].

In order to achieve high output performance for energy harvesting, researchers have chosen materials with a high negative charge. Among the commonly used materials are FEP, PTFE, and PVDF, which are readily available and possess these characteristics [82,104,105]. However, the F-bonding of the hydrophobic layer when in contact with liquid will absorb the ions with low electronegativity, which decreases the TENG performance [19,32]. The high-electrical-conductivity ions are the reason for the low triboelectric charge on the hydrophobic layer. Additionally, the adsorbed ions on the electrode layer will gradually reduce the transfer electron charge between the liquid and solid surface [106]. As shown in Figure 9a,c, the output voltage decreases when the ion concentration increases. With different types of ions (Figure 9b,d), the voltage also has different values. The results indicated that the output performance depends a lot on the properties of the ions, thereby promising that the L–S TENG device can be used for ions or chemical detection. The pH value and temperature of the water are also other affecting parameters of the TENG. The same as with the ion concentration, the positive hydrogen ion (H+) increase leads to a decrease in the output voltage, as shown in Figure 9e [13]. As explained in the working principle, the anions causing an imbalance in the charge of liquid molecules leads to the difficulty of the liquid interacting with the solid surface, which is harmful to the output performance of the TENG. As shown in Figure 9f, the short-circuit current density (Jsc) decreases with increases in water temperature. The results can be explained by the decreasing rate of change in the dielectric constant and the polarity of water with increasing temperature [43].

## 4. L–S TENGs as Self-Powered Active Pressure/Touch Sensors

Based on the above-mentioned operating principle of the L–S TENG, it is deduced that the variation in intensity or frequency of the dynamic fluid greatly affects the TENG electrical output; therefore, L–S TENGs can potentially appear in self-powered sensor applications in which they monitor the operations of industrial processes or detect human motions. In addition, the natural wetness of L–S TENGs provides decent benefits, such as self-cleaning and heat-resistant properties, so that they can achieve outstanding durability and stability for long-term operations, even under severe conditions. From these aspects, one of the potential applications of the L–S TENG is their use as active sensors for the monitoring of external pressure/touch in the operations of industrial processes or human motions. These triboelectric sensors with simple structures have been developed based on an L–S TENG, taking into account pressure, flow speed, human motions, fluctuation of a liquid surface, and so on [31,107,108,109,110,111,112,113].

### 4.1. L–S TENGs as Self-Powered Physical Sensors

It appears that L–S TENGs can offer various modes of human gesture recognition and environmental monitoring based on both the input mechanical energy and the spatial arrangement of liquid and solid materials. One example of this is a solid–liquid–solid mode TENG developed by Wang et al., which utilizes a device consisting of PTFE, water, and graphite to detect the amplitude and frequency of human fingers [114]. The water in the device not only serves as a triboelectric material but also transmits energy and signals, while the modified graphite electrode enhances the output of the TENG. By tapping a finger on the PTFE sheet, this device can convert the output voltage into international Morse code using spikes and flat band signals. Two methods of Morse code encoding based on amplitude and frequency are depicted in Figure 10a, with the size and duration of signals representing “dot” and “dash”. The receiving signals can be decoded in real-time on a computer screen or mobile phone.

Regarding environmental monitoring, researchers have also introduced several self-powered sensors-based L–S TENGs with remarkable performance. In particular, a self-powered water temperature sensor was proposed by Xiong et al. by applying the self-restoring, waterproof, tunable SMPU microstructural mats to fabricate the TENG with excellent superhydrophobicity (SCA = 152.2°) [115]. The self-restoring capability of the MS mats ensures a stable triboelectric performance and a long lifespan, even under impact and heating using hot water, thanks to the gradually increasing surface roughness during the structural recovery process triggered by water temperature. A single-electrode water-TENG design is depicted in Figure 10b, producing a series of output voltages under varying water temperatures and impacting times. Moreover, the recovery of surface roughness was recorded, and it was observed that higher temperatures could expedite the recovery process before complete recovery.

Zhang et al. built and researched a self-powered water level sensor based on an L–ST TENG as an example of how the L–S TENGs may be discovered in a real-time water level monitoring system (Figure 10c) [116]. The precision of the water level measurement used in this study, which is ten times greater than that of the conventional draft mark on the ship, was measured correctly and promptly for dynamic ship draft monitoring. By analyzing the peaks and valleys in the dVoc/dt signals, the water level was discovered. The peaks of the dVoc/dt signals correlate to different drafts (1, 3, 5, and 7 cm). Positive peaks in the dVoc/dt signals indicate an increase in the water level, whereas a fall in the water level results in a reduction in the dVoc/dt value. Such a phenomenon is brought on by the fact that the Voc increases/decreases more quickly as the water rises/falls in the electrode region while doing so more slowly in the blank region.

Other than the above-mentioned functions, by combining liquid–solid contact electrification with the Venturi tube structure, a Venturi-type triboelectric flow sensor (VTTFS) was fabricated [117]. The experimental flow measuring system is shown in Figure 10d. The results show that, when the flow is progressively increased from 95 L/min to 215 L/min, the overall trend rises, and when the flow is gradually decreased, the overall trend declines. The matching pulse counts exhibit a superb linear connection with the varying flow. The VTTFS was also evaluated in irregular flow conditions, and the electrical pulse signal generated shows a one-to-one relationship to the flow. The industrial production, pipeline transportation, measurement research, and even medical healthcare equipment industries are all potential markets for this flow-sensing method.

### 4.2. L–S TENGs as Self-Powered Pressure/Force Sensors

Pressure sensors are developing rapidly across the world, with a variety of applications such as in healthcare monitoring systems [109,110], game control, and soft robotics [111,118]. In numerous real-world situations, pressure sensors must operate with a basic model or without a power source. Therefore, developing these systems to be small, wireless, and maintenance-free is made possible via self-powered sensor-based triboelectrification. Pressure-sensor-based L–S TENGs have a straightforward design and are economical for detecting in vast regions and low-temperature conditions [112].

To develop the L–S TENG-based pressure sensor, several device structures have been designed. For example, a highly flexible microfluidic channel based on the TENG has been demonstrated by some researchers [112,113,119], and a simple U-tube based on the TENG has also shown good results for pressure sensing [20,108]. The U-shaped TENG fabrications from common materials in the industry and daily life have been explored based on Pascal’s law in order to gain a deeper quantitative grasp of the L–S TENG-based pressure sensor. Figure 11a illustrates the testing configuration for the U-shaped TENG as a model for dynamic pressure sensing using a tube constructed of FEP with an inner diameter of 20 mm [20]. When pressure is applied to the U-tube, the largest height difference between the two column fluid levels is Δ*H*. The acceleration (a), with respect to a constant vibration frequency, is completely dictated by the vibration amplitudes (x), and the transferred charge can be calculated using Equation (2):(2)$$Q_{SC}=2\pi \sigma r∆H$$
where *σ* is the surface tribo-charges density, and *r* is the diameter of the tube. From this equation, the short-circuit current at *t* is the time given by:(3)$$I_{SC}=2\pi \sigma r\frac{d∆H}{dt}=2\pi \sigma rV(t)$$
where *V*(*t*) is the water flow rate. As can be seem, the short-circuit current (ISC) is proportional to *V*(*t*) and Δ*H*. Here, the pressure applied (P) into the U-tube can be represented by the height difference in the liquid level, so the measured output performance is proportional to the pressure. As shown in Figure 11c,f, the relationship of open-circuit voltages and short-circuit current with pressure displayed good linear behavior (R^2^ = 0.998) under a low-pressure regime (from 0.16 kPa to 0.54 kPa). The result has shown that detailed information about the ambient mechanical motions can be monitored using the TENG device when combining the analysis of both the Voc and *I_sc_* measurements. Moreover, using a microfluidic channel design can shrink down the repeated system and achieve higher output performance compared to a U-tube structure. Due to its advantages of being battery-free, having a straightforward operating mechanism, and being inexpensive, self-powered sensors can be used in blood pressure monitoring, pulse waveform monitoring, electronic skin monitoring, and other applications.

## 5. L–S TENGs as Self-Powered Chemical/Environment Sensors

Chemical detection and environmental factor sensing are crucial for maintaining the quality of water or wastewater. The self-powered sensor which detects changes in the development of a liquid’s characteristics may use the influence of the liquid phase on the L–S TENG [19]. As the proportionate triboelectric charge density varies, the produced electrical signal in an L–S TENG fluctuates. Assuming all other factors remain constant, the chemical characteristics of the liquid may affect how the charge is absorbed by the solid surface, changing the triboelectric charge density. We have attempted to categorize the group of self-powered sensors in this part using various liquid properties, e.g., ion liquid concentration [13,120], organic concentration [89,121], chemical detection [122], and biological response [35,43].

Figure 12a shows the average output voltage values generated by various NaCl solution concentrations, from 0 to 0.75 M. When the NaCl concentration rises from 0.005 to 0.1M, it appears that the voltage output of the L–S TENG drops quickly. Various other studies also demonstrate the harmful effects of different ionic compounds (such as ZnCl_2_, KCl, CaCl_2_, KNO_3_, NaNO_3_, KOH, NaOH, and so on) on the output performance [13,19,70,123]. Moreover, based on the electronegativity properties of the cation or anion, the decline in the voltage shows a significant difference. The sensitivity of the self-powered TENG sensor with NaCl concentrations ranging from 0 M to 0.75 M is shown in Figure 12b. The output voltage ratio (Δ*V*/*V*) and the concentration of NaCl have a linear relationship, and the sensor exhibits great sensitivity in the range of 0.005 M to 0.1 M. The outcome supports the possibility of using an L–S TENG in a self-powered sensor with very low ion liquid concentration. Moreover, Figure 12c,d illustrates the sensor’s stated selectivity [106]. Several modifying agents can be used to alter the output voltage ratio of each ion liquid under the same testing conditions. A TENG sensor was modified with dithizone to enable the detection of Pb^2+^, and the results demonstrate that additional heavy metal ions have bigger output voltage ratios. When diphenylcarbazide is used as the modifying agent, the outcome is the same as Cr^3+^. It is clear that self-powered L–S TENG sensors offer excellent selectivity and sensitivity for measuring ion concentration.

Fermentation, biomedicine, and other chemical activities all heavily rely on organic liquids and gases. As a result, monitoring and managing the organic concentration is essential for optimizing industrial operations and raising the quality of life. For the detection of ethanol, formaldehyde, and glucose, a self-powered sensor based on a triboelectric nanogenerator device has recently gained a lot of interest [89,121]. A self-powered TENG sensor design for organic concentrations based on industry-standard PTFE filtering membranes is depicted in Figure 13a. Positive water charges will build up on the surface side that is in touch with the liquid and PTFE when the water is propelled via external mechanical vibration. An alternating current is produced by the water repeatedly making contact with and separating from the PTFE surface. The current of the TENG substantially reduces when an organic liquid with increasing concentration is used to replace water. It can appear that the formaldehyde and ethanol concentrations are measured using the TENG instrument. Furthermore, by investigating the charge-transfer process at liquid–solid interfaces, the L–S TENG has potential applications in the detection of surface properties of materials and chemical sensing [64,124,125]. For instance, Zhang et al. [64] conducted a study where they developed a self-powered droplet TENG that utilized spatially arranged electrodes. The purpose of this device was to serve as a tool for investigating the charge-transfer process at liquid–solid interfaces. The results demonstrated high sensitivity for chemical sensing applications, particularly in detecting variations in electrolyte concentrations within organic solutions and the composition ratios of mixed organic solutions. Additionally, the study revealed that molecules capable of forming intermolecular hydrogen bonds exhibited higher levels of transferred charges. This finding suggests that hydrogen bonding likely plays a significant role in influencing the charge transfer dynamics at the liquid–solid interface.

In addition to an application for self-powered sensors, chemical sensing has been developed in recent years, owing to life safety and industrial process controls. Figure 14a,c shows the working mechanism of a single-electrode L–S TENG (SELS-TENG)- and contact-separation L–S TENG (CSLS-TENG)-based sensor [122,126]. The working mechanism is also the same and has been explained in the working principal section. Based on different chemicals, the output signal shows a different result. In Figure 14b, the current of the SLES-TENG showed a positive peak, with the values mainly around 100 nA for alcohol and 99.7% detection. On the contrary, for 99.5% acetone, the current has a negative peak, and the value drops to −80 nA. Moreover, the current for sensing NaOH and NaCl also shows the difference when the current value is around +200 nA and −250 nA. In summary, the varied ions present in the liquid have an impact on the output current. The voltage and current of the self-powered sensor-based TENG are shown in excellent detail in Figure 14d. It is clear that the voltage and current characteristics or values will affect the detection and classification of different liquids.

## 6. Conclusions

Sensors have become an integral component of modern technology, enabling devices and systems to detect and respond to signals from the environment, humans, animals, and other sources. In order to power these devices without using external power sources, self-powered sensors have emerged as a possible approach. The potential for creating sophisticated sensing systems has increased with the rise of L–S TENGs as a significant technology in the field of self-powered sensors. One of the main benefits of the L–S TENG is its capacity to produce electricity from a variety of mechanical stimuli. This makes it suitable for use in a variety of sensing applications, including pressure sensing, touch detection, flow-rate monitoring, and chemical and biological detection. The device is also highly efficient at converting mechanical energy into electrical energy, making it an attractive option for powering small electronic devices. Another advantage of the L–S TENG is its low cost and environmental friendliness. The device can be fabricated using simple and inexpensive materials, making it accessible to a wide range of users. Additionally, the device does not rely on external power sources, reducing the environmental impact associated with battery disposal and replacement.

Despite its many advantages, there are still challenges associated with the use of the L–S TENG for sensing applications. One of the main challenges is the need to optimize the device for specific applications, as the performance of the device can vary depending on the materials and design used. Additionally, the device may be susceptible to wear and tear over time, which can reduce its efficiency and longevity.

The development of self-powered sensors based on the L–S TENG represents a significant advancement in sensing technology, despite the challenges faced. With their low cost, high stability, and ability to generate electricity from mechanical stimuli, these sensors have the potential to revolutionize a wide range of industries and applications, from healthcare and biomedicine to environmental monitoring and industrial automation. As research in this field continues to progress, it is likely that we will see new and innovative applications of L–S TENG-based sensors in the years to come. The future directions for self-powered sensors based on the L–S TENG principle involve several key areas, such as the advancement of flexible and wearable L–S TENGs for integration into textile and wearable devices, the exploration of novel liquid–solid interactions and materials to enhance performance, the integration of L–S TENG-based sensors into smart systems and IoT platforms, advancements in energy storage and management technologies for efficient energy utilization, and the development of sustainable and scalable manufacturing processes. These directions will pave the way for the further development and widespread adoption of L–S TENG-based sensors.

## Figures and Tables

**Figure 1 sensors-23-05888-f001:**
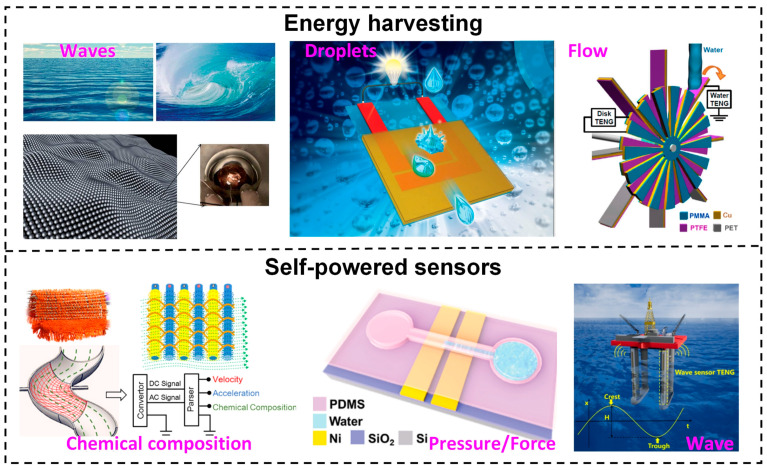
Application of the L–S TENG in energy harvesting and self-powered sensors. Reproduced with permission from [28], 2014, Royal Society of Chemistry; [29], 2014, Royal Society of Chemistry; [30], 2014, American Chemical Society; [31], 2018, American Chemical Society; [32], 2016, Wiley; and [33], 2019, Elsevier.

**Figure 2 sensors-23-05888-f002:**
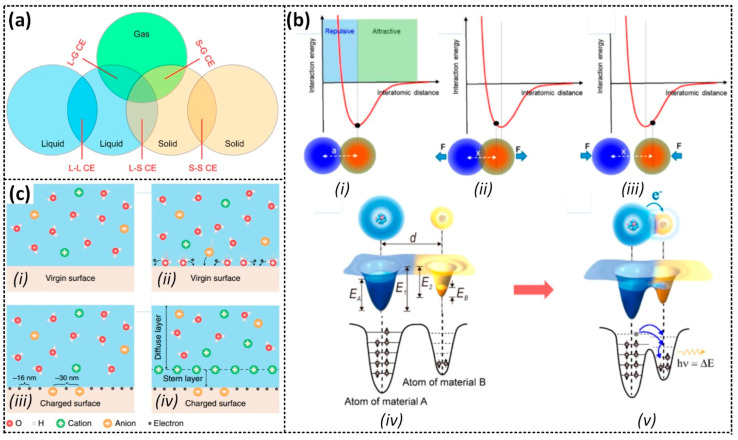
(**a**) Schematic of the contact electrification between different phases. (**b**) Interaction of two atoms at the (i) equilibrium position, (ii) repulsive region, and (iii) attractive region; (iv,v) electron transfer in contact. (**c**) Mechanism of contact electrification at the liquid–solid interface and formation of the EDL. Reproduced with permission from [40], 2022, American Chemical Society; and [41], 2020, Wiley.

**Figure 3 sensors-23-05888-f003:**
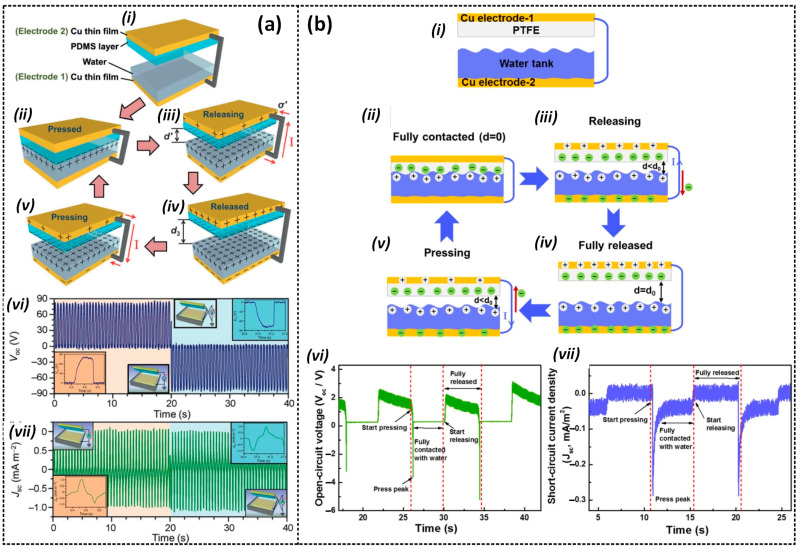
The working mechanism of the contact-separation mode: (i) Initial, (ii) Pressed. (iii) Releasing, (iv) Released, and (v) Pressing states, and the comparison of the measured results of the (vi) open-circuit voltage and (vii) short-circuit current densities of (**a**) the water-TENG proposed by Lin et al. [43] and (**b**) the LSTEG proposed by Yang et al. [46]. Reproduced with permission from [43], 2013, Wiley; and [46], 2018, Elsevier.

**Figure 4 sensors-23-05888-f004:**
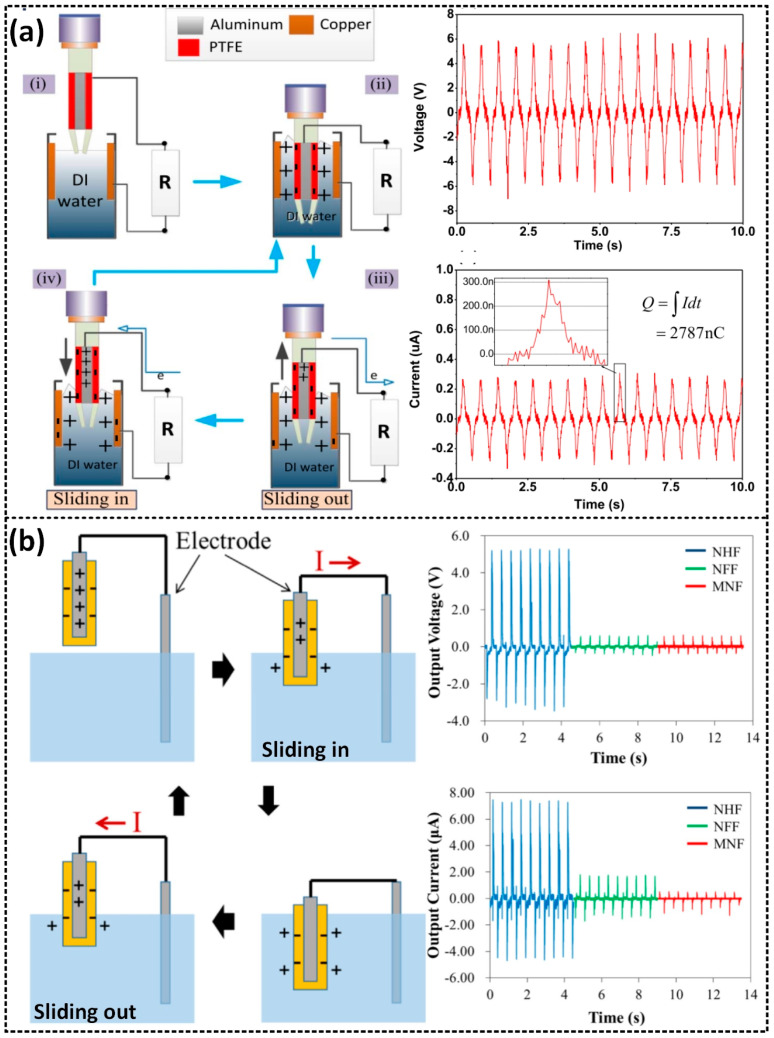
The working mechanism of the lateral sliding mode and the comparison of the measured results of the open-circuit voltage and short-circuit current of (**a**) the L-S FluTENG: (i) Fully separated, (ii) Fully contacted, (iii) Sliding out, and (iv) Sliding in states; and (**b**) the WTENG. Reproduced with permission from [48], 2017, Elsevier; and [47], 2018, Elsevier.

**Figure 5 sensors-23-05888-f005:**
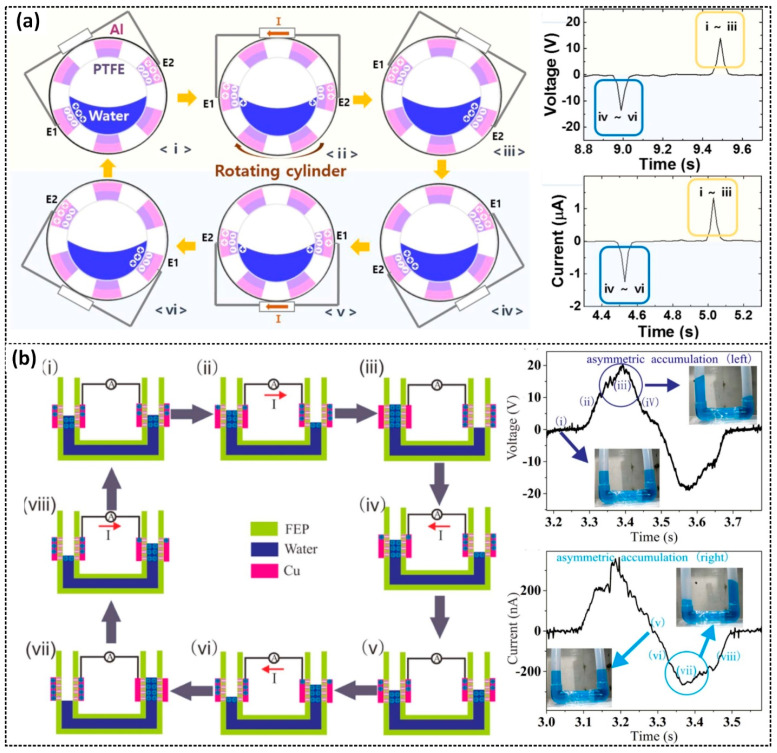
The working mechanism of the free-standing mode and typical voltage and current output signals of (**a**) the rotating water TENG: Generation of (i–iii) positive and (iv–vi) negative peak signals; and (**b**) the U-shaped TENG: Generation of (i–iv) positive and (v–viii) negative peak signals. Reproduced with permission from [56], 2016, Elsevier; and [20], 2017, Elsevier.

**Figure 6 sensors-23-05888-f006:**
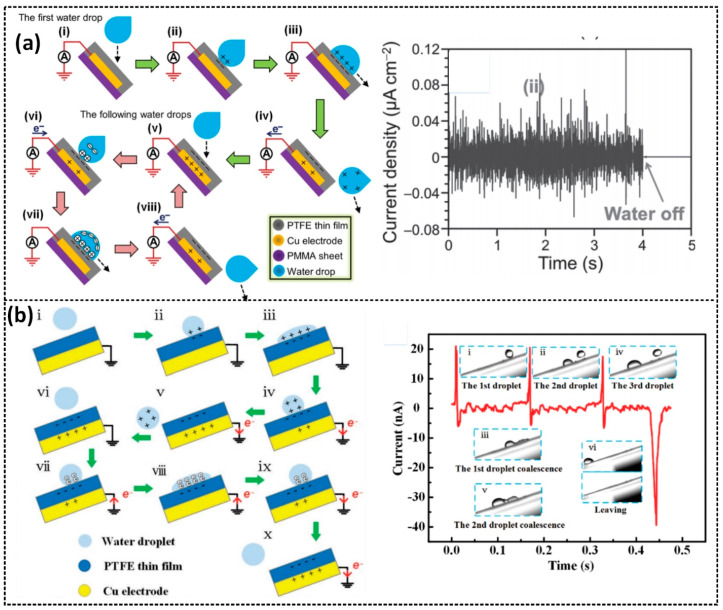
The working mechanism of the single-electrode mode and the typical output current signal of (**a**) a water-TENG with a superhydrophobic PTFE surface: (i) Precharged and (v) charged surfaces, water droplet in (ii & vi) contacting, (iii & vii) sliding, and (iv & viii) separated states with solid surface; and (**b**) the Wd-TENG: (i) Precharged and (vi) charged solid surfaces, water droplet in (ii & vii) contacting, (iii–iv & viii–ix) sliding, and (v & x) separated states with solid surface. Reproduced with permission from [15], 2014, Wiley; and [66], 2019, Wiley.

**Figure 7 sensors-23-05888-f007:**
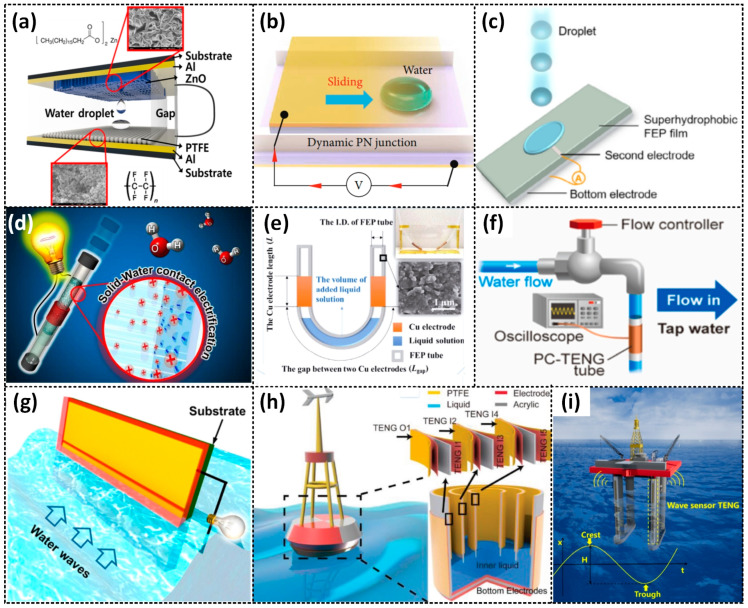
Schematic diagrams of (**a**–**c**) the droplet-based, (**d**–**f**) flow-based, and (**g**–**i**) wave-based L–S TENGs. Reproduced with permission from [45], 2019, Elsevier; [63], 2021, Science Partner Journals; [68], 2021, Wiley; [62], 2015, Springer Nature; [19], 2018, Springer Nature; [73], 2022, Elsevier; [51], 2014, American Chemical Society; [65], 2018, Wiley; and [33], 2018, Elsevier.

**Figure 8 sensors-23-05888-f008:**
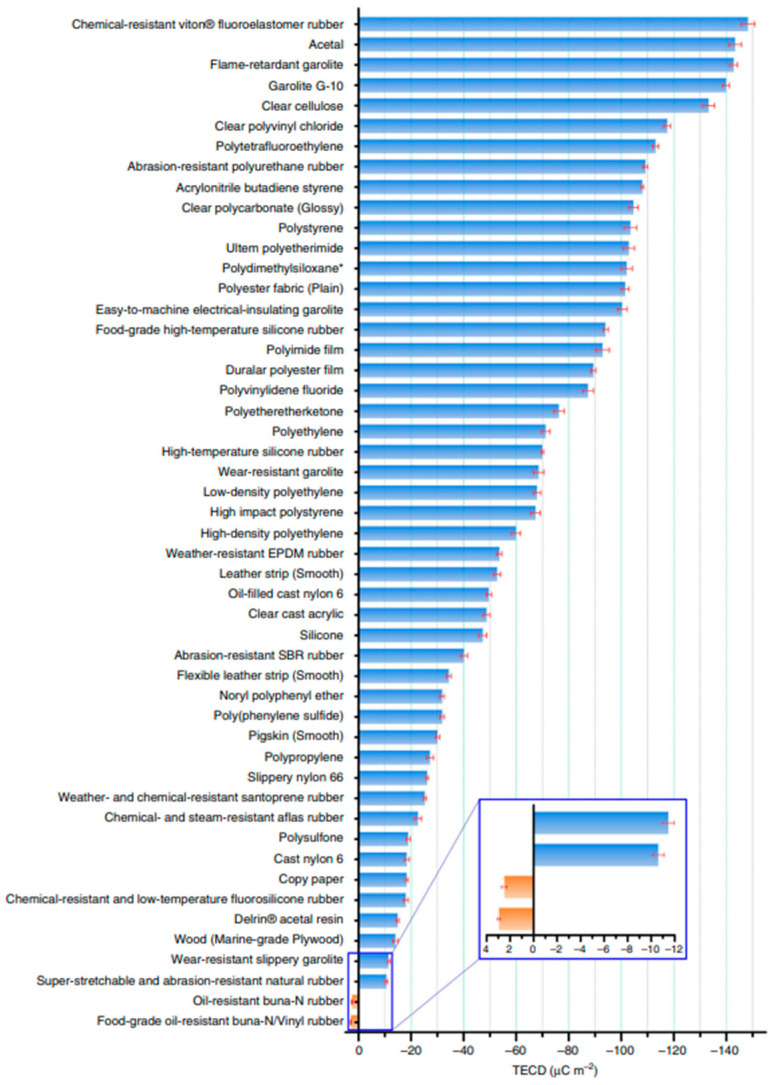
The triboelectric charge density (TECD) of different triboelectric materials. Reproduced with permission from [97], 2019, Nature Portfolio.

**Figure 9 sensors-23-05888-f009:**
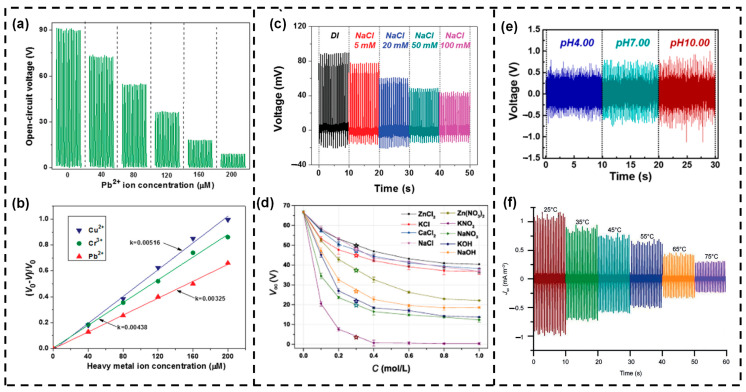
Performance characterization of the L–S TENG with different (**a**–**d**) properties of the ions, (**e**) pH values, and (**f**) temperatures of the water. Reproduced with permission from [106], 2016, Wiley; [32], 2016, Wiley; [106], 2016, Wiley; [19], 2018, Springer Science; [13], 2016, American Chemical Society; and [43], 2013, Wiley.

**Figure 10 sensors-23-05888-f010:**
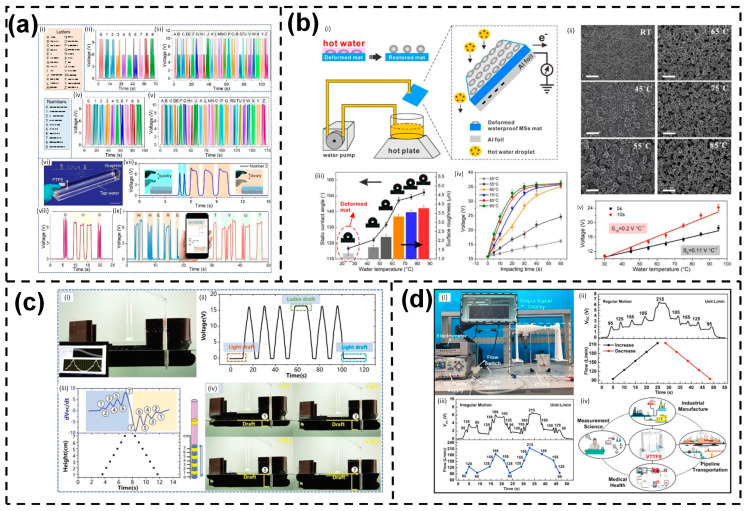
(a) Electrical response from the TENG and self-powered communication via solid–liquid–solid interactions. Inset images: (i) Morse code table, electrical responses based on magnitude and duration of the high level of (ii) & (iv) numbers and (iii) & (v) alphabets, respectively, (vi) Schematic diagram of the device, (vii) Signal waveform for Morse code’s transmission, Signal waveforms for the message (viii) “SOS” and (ix) “WHERE” and “TYUT”; (**b**) electrospun MS mat-based water-TENG for self-powered water temperature sensor. Inset images: (i) The setup and working mechanism, (ii) SEM images, (iii) The dependence of contact angle on water temperature, (iv) The dependence of output voltage on impacting time and water temperature, (v) Relationship between the output voltage and temperature; (**c**) the application of LST-TENG as a water level sensor for ship draft detecting. Inset images: (i) Experimental setup, (ii) Output voltage, (iii) The time variation of the output voltage with time and detected water level, (iv) The ship at different water level; (**d**) flow and level sensing via waveform-coupled liquid–solid contact electrification. Inset images: (i) Experimental setup, (ii) The dependence of the open-circuit voltage on (ii) regular flow and (iii) irregular flow, (iv) Application prospects of the device. Reproduced with permission from [114], 2022, Elsevier; [115], 2019, Elsevier; [116], 2019, Wiley; and [117], 2021, Elsevier.

**Figure 11 sensors-23-05888-f011:**
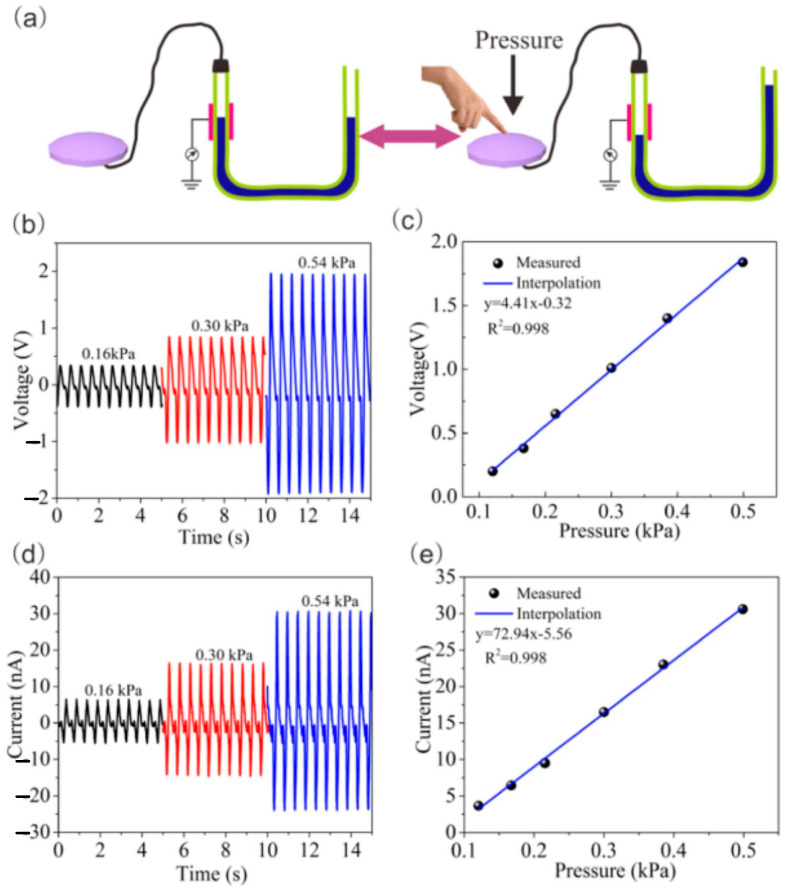
(**a**) Testing setup for the U-shaped TENG as a model for dynamic pressure sensing; (**b**) the open-circuit voltage peak at different pressures and (**c**) its relationship; (**d**) the short-circuit current peak at different pressures and (**e**) its relationship. Reproduced with permission from [20], 2017, Elsevier.

**Figure 12 sensors-23-05888-f012:**
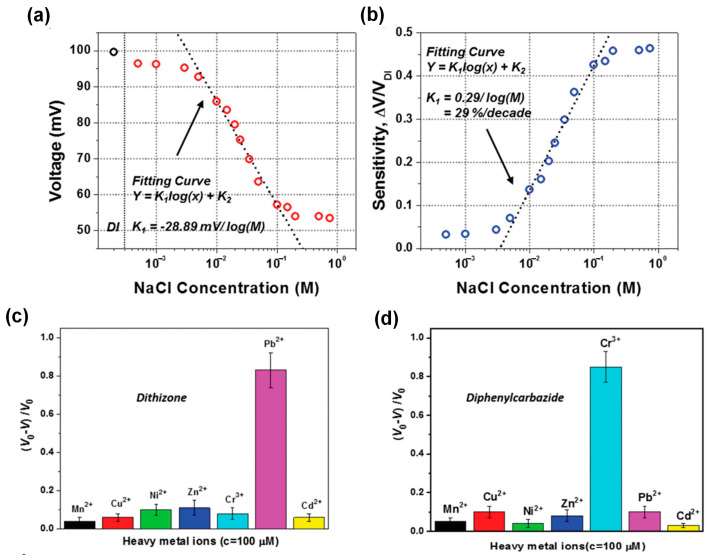
(**a**) Average output voltage values generated by various NaCl solution concentrations ranging from 0 to 0.75 M, and (**b**) the sensitivity of the sensor during its NaCl solution concentration. A test of the selectivity of the self-powered triboelectric sensor for (**c**) Pb^2+^ detection by using dithizone as the surface modifying agent; (**d**) Cr^3+^ detection by using diphenylcarbazide as the surface modifying agent. Reproduced with permission from [32], 2016, Wiley; and [106], 2016, Wiley.

**Figure 13 sensors-23-05888-f013:**
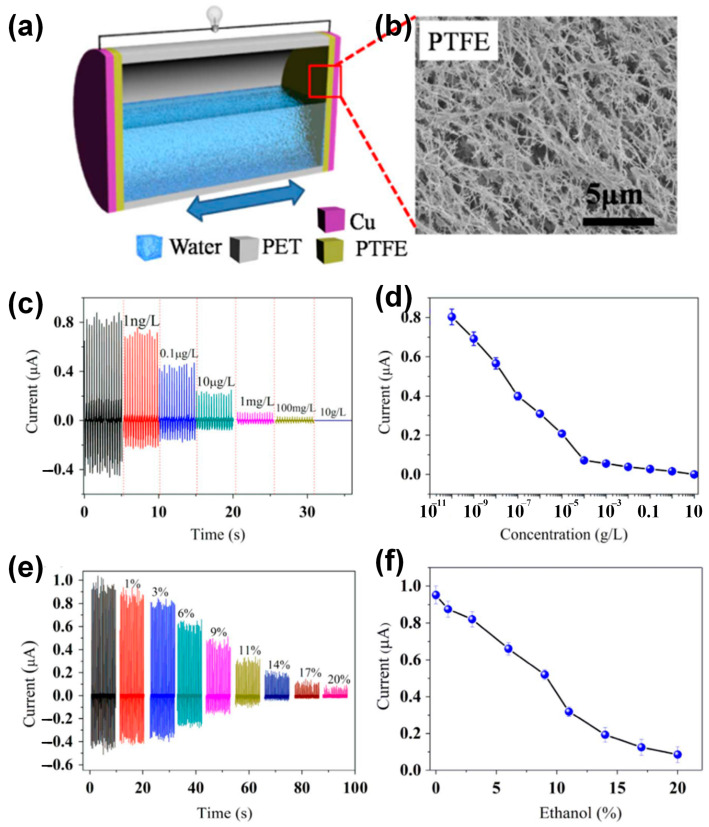
(**a**) Schematic illustration of the self-powered TENG sensor for organic concentrations and (**b**) the FESEM image of PTFE membrane surface; (**c**,**d**) short-circuit current for different formaldehyde concentrations; and (**e**,**f**) short-circuit current for different ethanol concentrations (percentage by volume). Reproduced with permission from [49], 2016, Elsevier.

**Figure 14 sensors-23-05888-f014:**
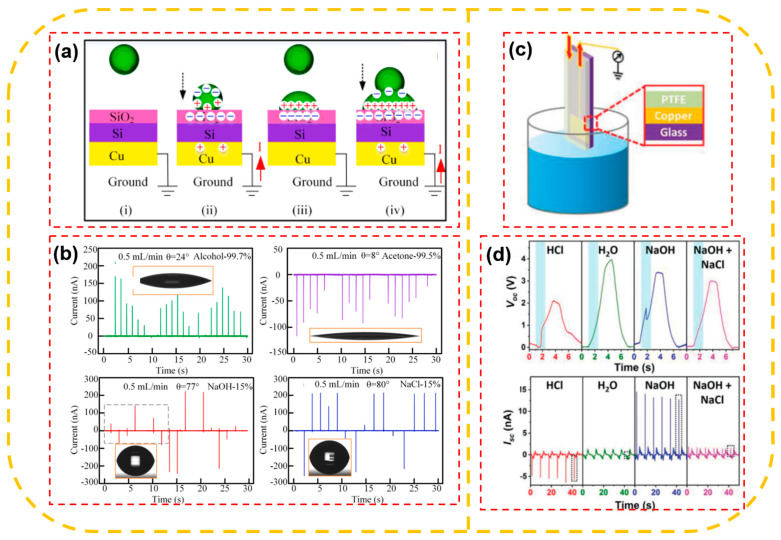
(**a**) The working mechanism of the L–S TENG-based sensor: (i) First droplet falls, (ii) Droplet contacts the surface, (iii) Second droplet falls, and (iv) Second droplet contacts the first droplet; and (**b**) Isc with different liquid alcohol, acetone, NaOH liquid, NaCl liquid values; (**c**) schematic illustration of the L–S TENG and (**d**) Voc and Isc in paraffin oil/water with different aqueous solutions of HCl (0.1 mol·L^−1^), deionized water, NaOH (0.1 mol·L^−1^), and mixture solution of NaOH and NaCl (0.1 mol·L^−1^). Reproduced with permission from [122], 2019, Elsevier; and [126], 2019, Wiley.

## Data Availability

Not applicable.

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
