# Peer review of "Recent Progress in Self-Powered Sensors Based on Liquid–Solid Triboelectric Nanogenerators"

_sensors, 2023, doi:10.3390/s23135888_

Round 1

Reviewer 1 Report

The authors comprehensively review recent progress in self-powered sensing based on L-S TENG, encompassing working mechanism, operating mode, performance-optimizing, and self-powered sensing paradigmatic examples. We think this manuscript is well-organized and can be published after addressing the following concerns.

1. Please reorganize the figures in the manuscript,  unify the format, and replace low-resolution figures.

2. In this manuscript, the authors paid more attention to summarizing the published works, instead of giving their own view on this area. We recommend the authors discuss more in the conclusion part. Talk more about the development of this area.

Author Response

Please see an attached file.

Reviewer 2 Report

This review outlines the working principle for liquid-solid contacts with three basic modes. The affecting parameters are also reviewed based on the properties of the liquid and solid phases. From reading this draft I recommend publication but strongly propose some restructuring to better present the subject to non-experts. I have provided some specific examples below.

1. Introduction: TENG is based on the contact electrification between materials, thus a better description of what contact electrification is needed here.

2. Section 3: Regarding the properties of water form effect on the amount and frequency of contact between liquid and solid surfaces, the droplet movement, for example, velocity should be included and detailed discussed, and related papers need to be cited, for example, Journal of Materials Chemistry A, 11,2023, 5696-5702.

3. Section 5: A recent paper developed a self-powered droplet-TENG with spatially arranged electrodes as a probe for measuring the charge transfer process between liquid and solid interfaces. By means of measuring the electric signal on spatially arranged electrodes, such type of droplet-TENG showed a high sensitivity to the ratio of solvents in the mixed organic solution (ACS nano, 2021, 15, 14830-14837). It would be better to discuss this literature in section 5.

4. Some literature related to the TENG probe for measuring the charge transfer at liquid-solid interface need to be included: ACS Nano 2023, 17, 2, 1646–1652; ACS Nano 2020, 14 (12), 17565−17573.

5. The figure caption of Figure 9 is confusing for me, only figure 9f is related to environment, the others are all concentration? Better to be rephrased.

6. I would suggest some editing for English.

I would suggest some editing for English.

Author Response

Please see an attached file.

Reviewer 3 Report

The authors have produced a timely and quite well-structured review. Nevertheless, I invite them to revise the manuscript with the help of a professional translator with experience in the field of electrical engineering, as the text needs to be polished.

Just some remarks:

The sentence at line 65 repeats the one at line 62.

Line  84: “… as a results of tribology … “ is meaningless; it should be either deleted or detailed.

Line 96: “… allows for electron transfer and static charges”. What does “static charges” mean? 

Line 437: “… change in the dielectric constant and polarity…” What kind of changes?

There are several other situations in which the electrostatic concepts are inappropriately employed. The authors should either ask the help of a physicist to check the manuscript thoroughly or to cite exactly the formulations made by the authors of the cited references.

Fig. 1 is nice, but the devices depicted in it should be named in the legend and – perhaps – described in the text. Otherwise, it is just a good-looking illustration, but which conveys no information…

I am not a native speaker of English, but I have no difficulty with reading a well-written text. In this manuscript, there are still too many sentences that are ambiguous, perhaps also because of the inappropriate usage of some electrostatic concepts.

Author Response

Please see an attached file.

Reviewer 4 Report

This review describes state-of-the art and progress in L-S-TEGs for self-powering IoT devices. It is informative and could be of great interest to the research community working on this important area. This review can be published after minor but mandatory revisions.

1. There a too many English style deficiencies. The paper must be thoroughly edited to correct the style.

2. Instead of lengthy description of the working mechanisms of the different devices (sometimes very difficult to understand) in the text, make detailed figure captions, referring to subfigures and describing the mechanisms there. this will help the reader understand them.

see above

Author Response

Please see an attached file.
